# Molecular Detection of Minimal Residual Disease before Allogeneic Stem Cell Transplantation Predicts a High Incidence of Early Relapse in Adult Patients with *NPM1* Positive Acute Myeloid Leukemia

**DOI:** 10.3390/cancers11101455

**Published:** 2019-09-28

**Authors:** Federico Lussana, Chiara Caprioli, Paola Stefanoni, Chiara Pavoni, Orietta Spinelli, Ksenija Buklijas, Anna Michelato, GianMaria Borleri, Alessandra Algarotti, Caterina Micò, Anna Grassi, Tamara Intermesoli, Alessandro Rambaldi

**Affiliations:** 1Hematology and Bone Marrow Transplant Unit, Azienda Socio Sanitaria Territoriale Papa Giovanni XXIII, 24127 Bergamo, Italy; chiaracaprioli@gmail.com (C.C.); pstefanoni@asst-pg23.it (P.S.); cpavoni@asst-pg23.it (C.P.); ospinelli@asst-pg23.it (O.S.); ksenija.buklijas@gmail.com (K.B.); annamick@yahoo.it (A.M.); gborleri@asst-pg23.it (G.B.); aalgarotti@asst-pg23.it (A.A.); cmico@asst-pg23.it (C.M.); agrassi@asst-pg23.it (A.G.); tintermesoli@asst-pg23.it (T.I.); arambaldi@asst-pg23.it (A.R.); 2Department of Oncology and Hematology, Universita’ degli Studi di Milano, 20122 Milano, Italy

**Keywords:** acute myeloid leukemia, nucleophosmin (*NPM1*), allogeneic stem cell transplantation

## Abstract

We analyzed the impact of alloHSCT in a single center cohort of 89 newly diagnosed *NPM1^mut^* AML patients, consecutively treated according to the Northern Italy Leukemia Group protocol 02/06 [NCT00495287]. After two consolidation cycles, the detection of measurable residual disease (MRD) by RQ-PCR was strongly associated with an inferior three-year overall survival (OS, 45% versus 84%, *p* = 0.001) and disease-free survival (DFS, 44% versus 76%, *p* = 0.006). In MRD-negative patients, post-remissional consolidation with alloHSCT did not provide a significant additional benefit over a conventional chemotherapy in terms of overall survival [OS, 89% (95% CI 71–100%) versus 81% (95% CI 64–100%), *p* = 0.59] and disease-free survival [DFS, 80% (95% CI 59–100%) versus 75% (95% CI 56–99%), *p* = 0.87]. On the contrary, in patients with persistent MRD positivity, the three-year OS and DFS were improved in patients receiving an alloHSCT compared to those allocated to conventional chemotherapy (OS, 52% versus 31%, *p* = 0.45 and DFS, 50% versus 17%, *p* = 0.31, respectively). However, in this group of patients, the benefit of alloHSCT was still hampered by a high incidence of leukemia relapse during the first year after transplantation (43%, 95% CI 25–60%). Consolidative alloHSCT improves outcomes compared to standard chemotherapy in patients with persistent *NPM1^mut^* MRD positivity, but in these high-risk patients, the significant incidence of leukemia relapse must be tackled by post-transplant preemptive treatments.

## 1. Introduction

Nucleophosmin (*NPM1*) is one of the most commonly mutated genes in acute myeloid leukemia (AML), being detectable in about 30% of de novo AML cases [1]. Acute myeloid leukemia with mutated *NPM1 (NPM1^mut^)* represents a distinct entity in the revised World Health Organization (WHO) classification, and the prognosis of patients carrying this mutation is generally considered favorable [2]. Accordingly, the European LeukemiaNet included *NPM1^mut^* patients without *FLT3-*internal tandem duplication (ITD) or with a concomitant *FLT3*-ITD mutation with an allelic ratio <0.5 and normal karyotype into a favorable prognostic group [2]. For these patients, in first complete remission (CR1), a post-remissional consolidation with allogeneic hematopoietic stem cell transplantation (alloHSCT) is usually not recommended. The detection of the *NPM1^mut^* gene represents a reliable marker to track measurable residual disease (MRD) by RT-PCR. Recent studies have shown a correlation between *NPM1^mut^* MRD and an adverse clinical outcome [3,4,5,6,7,8,9]. This association has generated substantial interest in using results of MRD testing for the decision of allocating patients to transplant, although the benefit associated with alloHSCT remains to be investigated, since only a small number of patients have been analyzed so far [4,10]. Other studies have shown that similar conclusions could be drawn when MRD is determined by multiparametric flow cytometry (MFC), regardless of molecular classification [11]. Since the outcome of *NPM1^mut^* MRD-positive patients is usually poor, the choice of an allogeneic transplant is usually considered for these patients, although this approach is not yet supported by prospective studies. For this reason, we analyzed the impact of allogeneic transplant in a cohort of *NPM1^mut^* AML patients consecutively treated according to the Northern Italy Leukemia Group (NILG) protocol 02/06 [ClinicalTrials.gov Identifier: NCT00495287]. In this study, patients were eligible to allogeneic transplant in case of leukocytosis (>50 × 10^9^/L) at diagnosis, the presence of *FLT3*-ITD mutation (no matter the allele burden), or the persistence of molecular MRD positivity after two consolidation cycles [12].

## 2. Results

### 2.1. Patients’ Characteristics

Of 89 adult patients (median age 54, range 16–73) with newly diagnosed *NPM1^mut^* AML, 84 (94%) achieved complete remission (CR) after the first cycle, and two patients achieved CR after two cycles of induction chemotherapy. Three patients did not achieve CR, and were excluded from this study. MRD status after consolidation chemotherapy was available for 72 patients (84%). It is worth noting that higher WBC and lactate dehydrogenase (LDH) at diagnosis were associated with the risk of the persistence of MRD positivity after two consolidation cycles, confirming that these parameters are adverse clinical characteristics. The main clinical findings of the analyzed patients are summarized in Table 1. 

Fifty-four patients received an alloHSCT in CR1, 41 (76%) received an alloHSCT after meyloablative, and 13 (24%) received an alloHSCT after a reduced intensity conditioning regimen. Donors were human leukocyte antigen (HLA)-identical siblings (*n* = 10), matched unrelated (*n* = 35), family mismatched (haploidentical, *n* = 3), or cord blood units (*n* = 6). The allogeneic graft source was represented by stem cells obtained from the bone marrow-derived stem cells (9%), G-CSF mobilized peripheral blood (80%), or cord blood units in the remaining 11% of patients.

### 2.2. Long-Term Outcomes

For the whole patients’ cohort (*n* = 89) with a median follow-up of three (range 0.5–11) years, the three-year overall survival (OS) was 62% (95% CI, 52–74%) and disease-free survival (DFS) was 51% (95% CI, 40–64%) (Figure 1A,B). The three-year OS and DFS were significantly different according to the MRD detected after two consolidation chemotherapy cycles. The OS was 45% (95% CI, 32–65%) in MRD positive versus 84% (95% CI, 71–99%) in MRD-negative patients, *p* = 0.001. Similarly, the DFS was 44% (95% CI, 28–64%) versus 76% (95% CI, 61–95%), *p* = 0.006], respectively (Figure 1C,D). In MRD-negative patients, post-remissional consolidation with alloHSCT did not provide a significant additional benefit over a conventional chemotherapy in terms of OS [81% (95% CI 64–100%) versus 89% (95% CI 71–100%), *p* = 0.59] and DFS [75% (95% CI 56–99%) versus 80% (95% CI 59–100%), *p* = 0.87] (Figure 2A,B). On the contrary, when a persistent MRD positivity was documented, the three-year OS and DFS were improved in patients receiving an alloHSCT compared to those allocated to conventional chemotherapy (OS, 52% versus 31%, *p* = 0.45 and DFS, 50% versus 17%, *p* = 0.31, respectively) (Figure 2C,D). However, in this group of patients, the benefit of alloHSCT was still largely hampered by a high incidence of leukemia relapse during the first year after transplantation (40%, 95% CI 24–56%) (Figure 3A). In contrast, the risk of non-relapse mortality (NRM) was not particularly high (Figure 3B).

Not surprisingly, different levels of molecular MRD positivity (negative (undetectable or ≤0.01%), low (<0.1% but >0.01%), and high (≥0.1%)) translated into an apparent different clinical outcome. Notably, the benefit gained by an alloHSCT was greater for patients with low pre-transplant MRD positivity (OS 83% versus 33%, *p* = 0.08; DFS 83% versus 38%, *p* = 0.02) compared to those undergoing transplantation with high levels of MRD positivity (OS 43% versus 30%, *p* = 0.65; DFS 42% versus 30%, *p* = 0.76). 

A sub-analysis among MRD-positive patients, who underwent alloHSCT, showed that the presence of FLT3-ITD mutation and a higher level of MRD positivity were significantly associated with an increased risk of relapse within one year after transplantation. In contrast, there were no significant differences in patients’ characteristics, such as age, white blood cell (WBC), and donor type between relapsed and not relapsed patients.

By multivariate analysis, the presence of *FLT3*-ITD mutation and the persistence of molecular MRD after consolidation chemotherapy were associated with a shorter OS and DFS, no matter the transplant consolidation (Table 2; Table 3).

## 3. Discussion

The analysis we present in this paper was undertaken to evaluate the clinical outcome according to *NPM1^mut^* MRD levels before transplantation. Points of strength of this study are represented by the prospective nature of the original trial (the Northern Italy Leukemia Group (NILG) protocol 02/06, ClinicalTrials.gov Identifier: NCT00495287), the single center transplant experience reported in this consecutive cohort of *NPM1^mut^* AML patients [12], and the prolonged follow-up period.

In keeping with a previous report, we confirmed a beneficial effect of alloHSCT in patients with persistent MRD positivity after consolidation chemotherapy [13]. The long-term follow-up of our analysis shows an improvement in terms of OS and DFS for patients with persistent MRD positivity undergoing alloHSCT compared with those not undergoing transplantation. In contrast, in the MRD-negative group, a post-remissional consolidation with alloHSCT or conventional chemotherapy was equally effective in terms of both OS and DFS. These results confirm the monitoring of *NPM1* MRD as a good marker to detect patients with a higher risk of adverse outcomes who might benefit from alloHSCT in CR1, thus enabling to spare this risky procedure in those who might be cured only with chemotherapy [4,7,9]. Although alloHSCT improves outcomes, the therapeutic benefit of alloHSCT in patients with persistent MRD positivity is only partial, which is mainly due to a high risk of relapse during the first year after alloHSCT. Thus, an effective prophylactic or preemptive therapy in post-transplantation might be important in order to reduce the rate of relapse [14,15,16,17,18]. Moreover, although the relatively small number of patients precludes drawing definitive conclusions, among patients who were MRD-positive after two consolidation cycles, we observed a negative effect of increasing levels of MRD on OS and DFS. Although a survival benefit has been reported also for patients with lower MRD clearance [19], two studies have shown the negative impact on outcome in patients with high MRD, independently from other variables, such as *FLT3-*ITD mutation, or age [10,20]. The retrospective design of most studies represents an obvious limit of all these studies, which are also highly heterogeneous in terms of selection criteria to transplant, time points for MRD assessment, cutoffs, and methods used for MRD evaluation. Consequently, this issue remains a matter of debate, but it is likely that these patients with high *NPM1^mut^* MRD levels could benefit from additional chemotherapy or innovative treatments, such as venetoclax or gemtuzumab ozogamicin, in order to obtain a better MRD clearance before alloHSCT [21,22,23,24,25].

By multivariate analysis, the other factor that had an influence on clinical outcome in our cohort of patients was the presence of an *FLT3-*internal tandem duplication *(FLT3-*ITD*)* mutation. In our experience, we can document an adverse outcome in patients with *NPM1^mut^* AML harboring either a low or high allele ratio of *FLT3*-ITD mutation compared to those without *FLT3*-ITD mutation. This observation, in line with that of a previous study [26], suggests a note of caution in considering the prognosis of *NPM1^mut^* AML patients with a low *FLT3*-ITD allele ratio favorable [2]. For this reason, when an appropriate donor is available (HLA identical sibling or a matched unrelated), we are keen to always consider alloHSCT in CR1 for *NPM1^mut^*, *FLT3*-ITD positive AML patients, no matter the allele burden of this latter mutation. This holds particularly true in young patients, with a relatively low risk of non-relapse mortality [27,28]. This position is also supported by the results of a retrospective study showing an advantage in survival when transplantation is performed in CR1 compared to CR2 [29], and by the very recent analysis of the European Society for Blood and Marrow Transplantation, showing that alloHSCT is associated with a lower risk of hematological relapse compared to chemotherapy in patients with isolated *NPM1^mut^* AML [30].

Some limitations of the current study need to be considered, including the absence of evidence regarding the most clinically significant time points and MRD thresholds to be considered, but also with respect to the correlation of MRD with other known prognostic indicators, such as coexisting molecular mutations [1]. In addition, in NPM1 mutated patients, we documented the presence of additional chromosomal abnormalities only in five patients, and this prevents us from evaluating the recently described, negative prognostic impact associated with this finding [31]. We believe that our data must be corroborated by further analysis performed on a larger group of patients. 

## 4. Materials and methods

### 4.1. Patients, Diagnosis, and Minimal Residual Disease Evaluations

From 2006, 89 newly diagnosed *NPM1^mut^* AML patients, consecutively treated according to the NILG protocol 02/06 (ClinicalTrials.gov Identifier: NCT00495287) were analyzed. Briefly, all participants received conventional induction chemotherapy with idarubicin, cytarabine, and etoposide, or a sequential high-dose cytarabine and idarubicin. Post-induction treatment included additional chemotherapy courses with high-dose cytarabine, while final consolidation was based on a study-specific risk stratification and comprised high-dose cytarabine courses, autologous transplant, or alloHSCT. Detailed treatment descriptions of the trial have been reported previously [12]. According to the study design, patients were considered eligible to allogeneic transplant in first remission if, at diagnosis, they were *FLT3*-ITD positive, or had a high white blood cell (WBC) count (>50 × 10^9^/L), or they showed a persistent MRD, as molecularly detected after consolidation with high-dose cytarabine. Molecular analysis of *NPM1* and *FLT3* status was performed in all patients at diagnosis. Molecular MRD monitoring of *NPM1* was determined in the bone marrow and peripheral blood by real-time quantitative polymerase chain reaction analysis (RQ-PCR) according to a validated method [32]. MRD levels were expressed as a percentage (ratio of the NPM1 copies to the housekeeping gene ABL copies × 100); the sensitivity level was 0.01%, and MRD positivity was defined as any level above 0.01%. Response to treatment and relapse were assessed according to International Working Group criteria [33]. The study was approved by the local Institutional Review Board, and it was conducted in accordance with the Declaration of Helsinki. All patients provided written informed consent. 

### 4.2. Study Endpoints and Statistical Methods

Molecular analysis of *NPM1* status was performed in all patients at diagnosis and during the consolidation phase with high-dose cytarabine courses. Patients who achieved molecular MRD levels below 0.01% after consolidation and before conditioning were defined as MRD-negative, while patients carriers of any other positive MRD level in the bone marrow or peripheral blood were defined as MRD-positive. The endpoints of the study were defined according to the standard criteria [33]. Overall survival (OS) was defined as the probability of survival irrespective of disease state at any point in time from diagnosis. Patients alive at their last follow-up were censored. Disease-free survival (DFS) was measured from the time of CR1 until relapse or death. Cumulative incidence of relapse (CIR) was measured from the time of CR1 until relapse. Relapse was defined by the recurrence of more than 5% of myeloblasts in the peripheral blood or in the bone marrow and/or by the presence of extramedullary disease.

Baseline continuous characteristics were presented as median with range and compared, between consolidation with or without alloHSCT, using the Mann–Whitney U-test. Categorical variables were reported with absolute and percentage frequencies and compared with the chi-squared test or Fisher’s exact test. OS and DFS were estimated by the Kaplan–Meier method, and any differences in MRD and consolidation groups were evaluated with the log rank test. CIR was estimated by the cumulative incidence function, considering death as a competing event. Cox models were used to estimate hazard ratios with 95% confidence intervals (CI) in univariate and multivariate analysis on survival outcomes; in this context, alloHSCT was considered as a time-dependent variable. All reported P values are two-sided, and a 5% significance level was fixed. All analyses were performed with R software, version 3.5.0.

## 5. Conclusions

In conclusion, our study shows that consolidative alloHSCT improves OS and DFS compared to standard chemotherapy in patients at higher risk of leukemia relapse due to the persistence of *NPM1^mut^* MRD positivity. However, MRD-positive patients remain at high risk for relapse during the first year after alloHSCT. These findings suggest that clinical studies evaluating experimental preemptive treatments after transplantation are warranted for improving disease-free and overall survival in these high-risk patients.

## Figures and Tables

**Figure 1 cancers-11-01455-f001:**
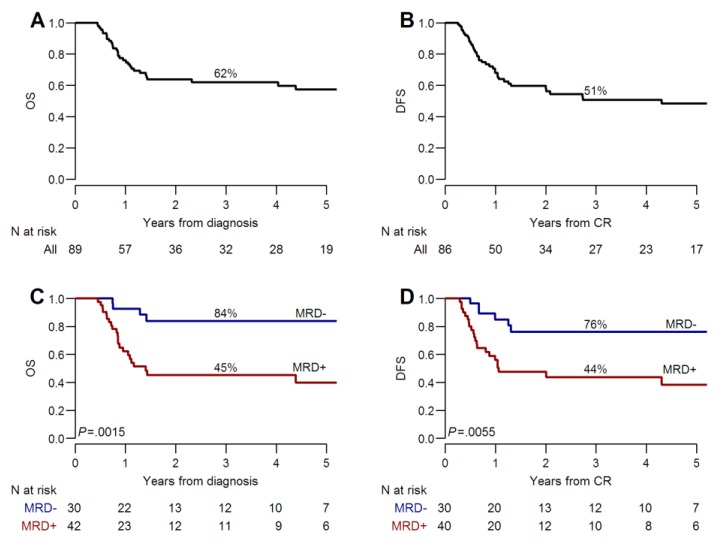
Overall survival and disease-free survival for the whole cohort (**A** and **B**) and according to MRD status (**C** and **D**).

**Figure 2 cancers-11-01455-f002:**
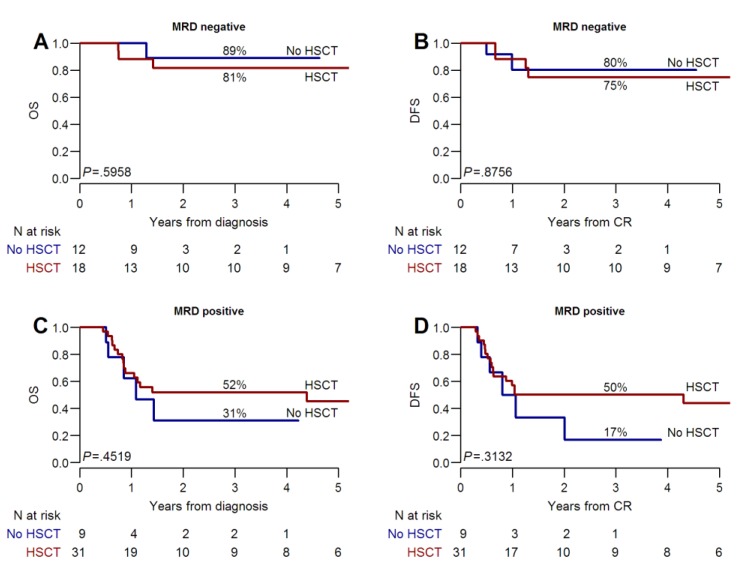
Overall survival and disease free survival according to alloHSCT. (**A**) and (**B**): patients with negative MRD after consolidation chemotherapy; (**C**) and (**D**): patients with positive MRD after consolidation chemotherapy.

**Figure 3 cancers-11-01455-f003:**
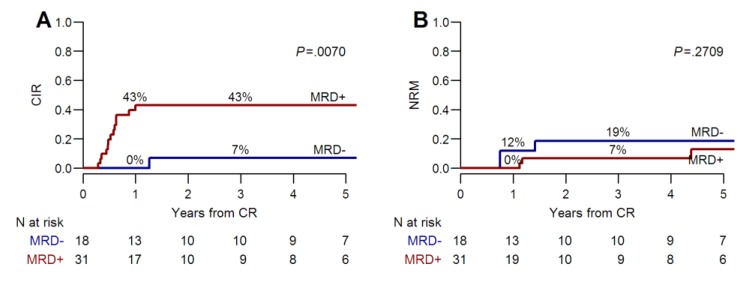
Cumulative incidence of hematologic relapse (**A**) and transplant related mortality (**B**) by MRD group in patients undergoing alloHSCT.

**Table 1 cancers-11-01455-t001:** Demographic and clinical patients’ characteristics.

Characteristics	All Patients *N* = 89	MRD Negative, *N* = 30	MRD Positive, *N* = 42	*p*^
Age, median (range)	54 (16–73)	52.5 (19–68)	54.5 (22–68)	0.97
Sex, N (%)				0.55
Male	41 (46.1)	15 (50)	18 (42.9)	
Female	48 (53.9)	15 (50)	24 (57.1)	
LDH U/L, median (range)	1124.5 (301–6000)	819 (355–2550)	1371 (351–4382)	0.06
WBC (×10^9^/L), median (range)	33.1 (1.2–262.9)	17.4 (1.3–180)	73 (2–262.9)	0.0004
Hemoglobin (g/dL), median (range)	8.9 (0.4–13.9)	8 (3.3–13.9)	9.3 (5.7–13.8)	0.07
Platelets (×10^9^/L), median (range)	51 (5–698)	46 (6–393)	53 (13–698)	0.47
Cytogenetic				0.30
Normal karyotype	84 (94.4)	27 (90.0)	41 (97.6)	
Abnormal °	5 (5.6)	3 (10.0)	1 (2.4)	
*FLT3-ITD*, N (%)				1.00
Negative	56 (66.7)	19 (65.5)	24 (63.2)	
Positive§, allelic ratio <0.5	10 (11.9)	3 (10.3)	4 (10.5)	
Positive§, allelic ratio ≥0.5	18 (21.4)	7 (24.1)	10 (26.3)	
MRD post consolidation ^#^, N (%)				-
Negative	30 (41.7)	30 (41.7)	-	
Positive ≤0.1	10 (13.9)	-	10 (23.8)	
Positive >0.1	32 (44.4)	-	32 (76.2)	
Consolidation				0.11
No alloHSCT	32 (37.2)	12 (40)	9 (22.5)	
AlloHSCT *	54 (62.8)	18 (60)	31 (77.5)	
Donor type, N (%)				0.31
Sibling	10 (18.5)	1 (5.6)	8 (25.8)	
Unrelated	35 (64.8)	15 (83.3)	17 (48.4)	
Cord Blood	6 (11.1)	1 (5.6)	4 (12.9)	
Haploidentical	3 (5.6)	1 (5.6)	2 (6.5)	

° Abnormalities include +8, t(2;13), inv3, i(7q). § Five patients have FLT3-ITD^+^ with an unknown allelic ratio. ^#^ Available for 72 (81%) out of 89 patients. * Three patients were excluded because underwent alloHSCT not in CR1. ^ MRD negative vs. MRD positive. AlloHSCT: allogeneic hematopoietic stem cell transplantation, ITD: internal tandem duplication, MRD: measurable residual disease, WBC: white blood cell, LDH: lactate dehydrogenase.

**Table 2 cancers-11-01455-t002:** Univariate and multivariable analysis for overall survival among 86 patients in complete remission (CR) after induction.

Factors	Univariate		Multivariable	
	HR (95% CI)	*p*	HR (95% CI)	*p*
Consolidation ^#^				
With AlloHSCT	1.00		1.00	
Without AlloHSCT	1.31 (0.59–2.89)	0.51	0.31 (0.08–1.19)	0.08
Age (years)	1.02 (0.99–1.05)	0.24	1.03 (0.97–1.1)	0.33
WBC (×10^9^/L)	1.01 (1–1.01)	0.03	1.00 (0.99–1.01)	0.98
FLT3-ITD				
Negative	1.00		1.00	
Positive, allelic ratio < 0.5	10.95 (3.9–30.71)	<0.0001	13.53 (2.87–63.78)	0.001
Positive, allelic ratio ≥ 0.5	8.19 (3.15–21.29)	<0.0001	13.29 (3.33–53.01)	0.0002
MRD post-consolidation *				
Negative	1.00		1.00	
Positive ≤ 0.1	2.21 (0.49–9.88)	0.30	12.55 (1.83–86.11)	0.01
Positive > 0.1	5.6 (1.87–16.79)	0.002	6.54 (1.71–25.04)	0.006

^#^ Time-dependent variable. * Available for 72 out of 86 patients.

**Table 3 cancers-11-01455-t003:** Univariate and multivariable analysis for disease-free survival among 86 patients in CR after induction.

Factors	Univariate		Multivariable	
	HR (95% CI)	*p*	HR (95% CI)	*p*
Consolidation ^#^				
With AlloHSCT	1.00		1.00	
Without AlloHSCT	0.75 (0.39–1.43)	0.38	0.25 (0.07–0.86)	0.03
Age (years)	1.01 (0.98–1.04)	0.37	1.00 (0.95–1.06)	0.89
WBC (×10^9^/L)	1.01 (1.00–1.01)	0.03	1.00 (1.00–1.01)	0.27
FLT3-ITD				
Negative	1.00		1.00	
Positive, allelic ratio < 0.5	5.52 (2.22–13.73)	0.0002	8.11 (2–32.84)	0.003
Positive, allelic ratio ≥ 0.5	5.07 (2.27–11.3)	0.0001	11.18 (3.32–37.7)	0.0001
MRD post-consolidation *				
Negative	1.00		1.00	
Positive ≤ 0.1	1.98 (0.56–7.01)	0.29	8.71 (1.49–50.88)	0.02
Positive > 0.1	4.04 (1.59–10.28)	0.003	5.66 (1.68–19.04)	0.005

^#^ Time-dependent variable. * Available for 72 out of 86 patients.

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
