# Peer review of "Molecular Detection of Minimal Residual Disease before Allogeneic Stem Cell Transplantation Predicts a High Incidence of Early Relapse in Adult Patients with NPM1 Positive Acute Myeloid Leukemia"

_cancers, 2019, doi:10.3390/cancers11101455_

Round 1
Reviewer 1 Report
Lussana et al present an interesting prospective study analyzing the outcomes of a small cohort of NPM1mut AML patient outcomes as associated with HSCT. This is a valuable study as designed, especially given that previous work may suggest a benefit to HSCT in some patients, however, HSCT alone carries risk factors, and clear clinical guidelines on which patients wherein the rewards would outweigh the risks are still unclear. This study is well designed given the limits of the patient population and is informative both for further research underlying AML mechanisms and for clinical guidelines for treatment.
I have a few clarification points that should be addressed to improve interpretation of the methods and overall study.
--for the non-clinical reader, a little more time in the introduction outlining the previous work for this study cohort (i.e. which consolidation cycle conditions and how were these judged) would be useful.
--please clarify the direct groups used for comparison to determine p values in the tables. For example, Table 1, is this comparing MRD neg to MRD pos? or one of these to the whole patient population. Same for Tables 2 and 3.
--there are several characteristics in Table 1 (LDH, WBC, hemoglobin) that are not discussed. While these are not statistically significant, they do have greater statistical significance than parameters the authors did discuss. Why are these considered unimportant?
--In the section starting on line 88, the first sentence here appears to be comparing without allHSCT vs with alloHSCT, while the second compares with alloHSCT vs without alloHSCT (as based on how the percentages are listed. This is confusing as written and should be kept consistent.
--given the patient number limitations it is understandable why some additional analyses can't be run. However, I am curious if the authors can analyze or speculate on the following: when comparing groups of patients, the MRD low cohort has only 10 patients (line 107), while the MRD high cohort has 30. The MRD high cohort has a 43% risk of relapse (Figure 3A). This suggests that if the MRD high cohort was separated into two sub-groups (those relapsing in the first year (~13 patients) and those not relapsing (~17 patients)), that there would still be sufficient power to compare these groups. This may allow a further determination of whether alloHSCT is beneficial when relapse is treated (as discussed on lines 135-138).
Overall I feel this is a sound study and addresses an appropriate question in this patient cohort.
Author Response
Reviewer 1
I have a few clarification points that should be addressed to improve interpretation of the methods and overall study.
1) for the non-clinical reader, a little more time in the introduction outlining the previous work for this study cohort (i.e. which consolidation cycle conditions and how were these judged) would be useful.
RESPONSE
We agree with the reviewer and as suggested we added additional information in the results of the revised version of the manuscript (page 8, lines 196-201)
2) please clarify the direct groups used for comparison to determine p values in the tables. For example, Table 1, is this comparing MRD neg to MRD pos? or one of these to the whole patient population. Same for Tables 2 and 3.
RESPONSE
We thank the reviewer for his/her suggestions. The requested modifications have been made in the tables of the revised version of the manuscript.
3) there are several characteristics in Table 1 (LDH, WBC, hemoglobin) that are not discussed. While these are not statistically significant, they do have greater statistical significance than parameters the authors did discuss. Why are these considered unimportant?
RESPONSE
We agree with the reviewer and we have included a new sentence to describe these differences (page 2, lines 70-73)
4) In the section starting on line 88, the first sentence here appears to be comparing without alloHSCT vs with alloHSCT, while the second compares with alloHSCT vs without alloHSCT (as based on how the percentages are listed. This is confusing as written and should be kept consistent.
RESPONSE
We thank the reviewer for pointing out this inconsistency which has been amended in the revised manuscript (page 3, lines 96-99).
5) given the patient number limitations it is understandable why some additional analyses can't be run. However, I am curious if the authors can analyze or speculate on the following: when comparing groups of patients, the MRD low cohort has only 10 patients (line 107), while the MRD high cohort has 30. The MRD high cohort has a 43% risk of relapse (Figure 3A). This suggests that if the MRD high cohort was separated into two sub-groups (those relapsing in the first year (~13 patients) and those not relapsing (~17 patients)), that there would still be sufficient power to compare these groups. This may allow a further determination of whether alloHSCT is beneficial when relapse is treated (as discussed on lines 135-138).
RESPONSE
We agree with the reviewer and accordingly, we detailed and analyzed the characteristics of these groups of patients in the revised version of the Results (page 6 from line 121 to line 125).
Reviewer 2 Report
The manuscript “Molecular detection of minimal residual disease before allogeneic stem cell transplantation predicts a high incidence of early relapse in adult patients with NPM1 positive acute myeloid leukemia” reports a study to discuss the relationship among minimal/measurable residual disease (MRD), 3-years overall survival (OS) and disease free survival (DFS) in acute myeloid leukemia patients with or without NPM1 mutation. Authors done an interesting work and paid much effort in data analysis, but the novelty of the research described in the manuscript is limited.
comments:
1. When we say that patients are "cancer free", that means no relapse over five years. So, authors should trace all data for five years, not 4 years only.
Author Response
When we say that patients are "cancer free", that means no relapse over five years. So, authors should trace all data for five years, not 4 years only.RESPONSE
We agree with the reviewer and accordingly we amended the figures in the revised version of the manuscript. Moreover we specified the median follow-up in the results section (page 3, line 89).
Reviewer 3 Report
In this paper, the authors revealed that the consolidated allo-HSCT improves OS and DFS compared to standard chemotherapy in patients at higher risk of leukemia relapse due to the persistence of NPM1mutMRD positivity. The MRD positive patients remain at high risk for relapse during the first year after allo-HSCT. This paper is the first report of the benefit of allo-HSCT was hampered by a high incidence of leukemia relapse during the first year after transplantation in patients with persistent MRD positivity receiving an allo-HSCT by analysis with cases allo-HSCT in a single center cohort of 89 newly diagnosed NPM1mutAML. The novel knowledge in this paper is little but important to the clinical matter and it influences on future's modified treatment against cases with NPM1mutAML. However it should be revised in data and descriptions. Criticisms regarding this revised paper are discussed below.

Comments

The graphs of OS and DFS in Fig 1-C and D, Fig 2-A, B, C and D, Fig 3-A and B may be seen simplified rough lines. Please refine to in-depth and detailed lines. Red dotted lines should be corrected in red solid lines. Please mention about a relation or correlation between NPM1mutmRNAand WT-1 mRNA in 89 cases. Please describe karyotype abnormalities in 89 cases and discuss the correlation with clinical outcomes. Recently it was reported that the karyotype abnormalities are significantly associated with outcome in NPM1mut/FLT3-ITDneg/lowAML (J Clin Oncol. 2019 Aug 20:JCO1900416. doi: 10.1200/JCO.19.00416).Author Response
Reviewer 3
1) The graphs of OS and DFS in Fig 1-C and D, Fig 2-A, B, C and D, Fig 3-A and B may be seen simplified rough lines. Please refine to in-depth and detailed lines. Red dotted lines should be corrected in red solid lines.
RESPONSE
As requested, the figures have been amended.
2) Please mention about a relation or correlation between NPM1mutmRNAand WT-1 mRNA in 89 cases.
RESPONSE
We do not have data to perform a correlation between NPM1 mut and WT-1 mRNA, because we do not use WT1 as MRD marker in keeping with ELN recommendation (Schuurhuis GJ et al. Blood.2018).
3) Please describe karyotype abnormalities in 89 cases and discuss the correlation with clinical outcomes. Recently it was reported that the karyotype abnormalities are significantly associated with outcome in NPM1mut/FLT3-ITDneg/lowAML (J Clin Oncol. 2019 Aug 20:JCO1900416. doi: 10.1200/JCO.19.00416).
RESPONSE
The reviewer poses a very interesting question that karyotype abnormalities may affect outcome in NPM1mut AML. We added and commented this important reference in the revised discussion of our manuscript (reference 31 and page 8 from line 188 to line 191). However, only 5 patients with abnormal karyotype were present in our study and such a small number prevents performing any further analysis. Taking advantage from the referee’s suggestion we added to the revised table 1, the few additional chromosomal abnormalities detected in our NPM1 mutated patients.
Round 2
Reviewer 2 Report
none